# Differential Marker Expression between Keratinocyte Stem Cells and Their Progeny Generated from a Single Colony

**DOI:** 10.3390/ijms221910810

**Published:** 2021-10-06

**Authors:** Dema Ali, Dana Alhattab, Hanan Jafar, Malak Alzubide, Nour Sharar, Salwa Bdour, Abdalla Awidi

**Affiliations:** 1Cell Therapy Center, The University of Jordan, Amman 11942, Jordan; demah.adnan@gmail.com (D.A.); dana.alhattab@kaust.edu.sa (D.A.); hanan.jafar@gmail.com (H.J.); malakalzubide@gmail.com (M.A.); nour.sharar@gmail.com (N.S.); 2Department of Biological Sciences, Faculty of Science, The University of Jordan, Amman 11942, Jordan; 3Laboratory for Nanomedicine, King Abdullah University of Science and Technology, Thuwal 23955-6900, Saudi Arabia; 4Department of Anatomy and Histology, School of Medicine, The University of Jordan, Amman 11942, Jordan; 5Department of Clinical Laboratory Sciences, Faculty of Science, The University of Jordan, Amman 11942, Jordan; 6Department of Hematology and Oncology, Faculty of Medicine, The University of Jordan, Amman 11942, Jordan

**Keywords:** keratinocyte stem cells, stem cells, transit amplifying cells, differentiated cells, holoclone, meroclone, paraclone

## Abstract

The stemness in keratinocyte stem cells (KSCs) is determined by their gene expression patterns. KSCs are crucial in maintaining epidermal homeostasis and wound repair and are widely used candidates for therapeutic applications. Although several studies have reported their positive identifiers, unique biomarkers for KSCs remain elusive. Here, we aim to identify potential candidate stem cell markers. Human epidermal keratinocytes (HEKs) from neonatal foreskin tissues were isolated and cultured. Single-cell clonal analysis identified and characterized three types of cells: KSCs (holoclones), transient amplifying cells (TACs; meroclones), and differentiated cells (DSCs; paraclones). The clonogenic potential of KSCs demonstrated the highest proliferation potential of KSCs, followed by TACs and DSCs, respectively. Whole-transcriptome analysis using microarray technology unraveled the molecular signatures of these cells. These results were validated by quantitative real-time polymerase chain reaction and flow cytometry analysis. A total of 301 signature upregulated and 149 downregulated differentially expressed genes (DEGs) were identified in the KSCs, compared to TACs and DSCs. Furthermore, DEG analyses revealed new sets of genes related to cell proliferation, cell adhesion, surface makers, and regulatory factors. In conclusion, this study provides a useful source of information for the identification of potential SC-specific candidate markers.

## 1. Introduction

The epidermal skin layer is an important protective barrier that is necessary for the maintenance of survival. Several conditions, including physical trauma, burns, and genetic diseases, affect this layer leading to compromised protective function. Attempts to restore this layer necessitate reconstructing it in a manner that preserves its continued existence. In other words, identifying, isolating, and propagating the “true” stem cell progenitors of this layer is essential for proper regeneration of the skin epidermis.

Efforts to identify progenitor cells of the skin epidermis revealed that the basal layer of the epidermis retains two cell populations with different proliferative capacity: (i) “keratinocyte stem cells (KSCs)” with high self-renewal potential; and (ii) “transit amplifying cells (TACs)” with high frequency of splitting when needed [1]. The majority of keratinocytes within the suprabasal layers of the epidermis are called “differentiated cells (DFCs),” which do not divide under normal conditions [2]. KSCs and their differentiated progeny (TACs and DFCs) can be identified either in vivo by label retention or in vitro by clonal analysis [3].

In further studies into the clonogenicity of KSCs and TACs, three clonal types of keratinocytes (termed holoclones, meroclones, and paraclones) with different proliferation capacities have been characterized using single-cell-based clonal analysis [4]. 

The holoclone, generated by the KSCs, demonstrates a high proliferative capacity, whereas the paraclone generated by TACs has very limited mitotic potential and usually gives rise to aborted colonies in vitro. The meroclone has an intermediate proliferation capacity and is regarded as a reservoir of TACs [5,6,7]. The transition from holoclones to meroclones and then to paraclones happens during a molecular mechanism called clonal conversion, and ultimately leads to cell differentiation [8,9]. 

Although identified in culture through single cell clonal analysis, unique molecular markers that differentiate these clonal cells are yet to be found, and therefore, the quest for reliable markers of human KSCs remains unresolved.

Performing a detailed transcriptome analysis can provide important information to better understand the biology and ontogeny of each cell type. Several studies using different cell surface markers from different sources and different culturing conditions have analyzed the gene expression profile in KSCs, TACs, and DFCs [10,11,12,13]. Although distinct data filtering strategies and various degrees of statistical stringencies were applied in these studies, they have introduced many variables in analyzing the whole gene expression profile, resulting in varied and sometimes conflicting results. Therefore, we hypothesized that comparing the gene expression profiles of the cultured holo-, mero-, and paraclones based on single-cell clonal analysis could provide better insights into the differential molecular signatures of the KSCs, TACs, and DFCs. 

In this study intending to identify candidate stem cell markers, we analyze the comparative clonogenic potential and molecular characteristics of single-cell-derived cultured holo-, mero-, and paraclone colonies of human epidermal keratinocytes (HEKs).

## 2. Results

### 2.1. Isolation and Characterization of Cultured Human Epidermal Keratinocytes

An average of 2.4 × 10^4^ cells/mm^2^ was obtained from each neonatal foreskin sample. After 9–12 days of primary culture (80–90% confluency), the HEKs formed colonies of tightly packed cohesive cells with a typical epithelial cell morphology (Figure 1a). High expression levels of keratins 14 and 5 (K14/K5) were detected in the immunostained HEKs (Figure 1b). These types of keratins are considered markers for undifferentiated cells in the basal layer of the epidermis. The CFE assay used to evaluate the clonogenic potential of cultured HEKs revealed that the ability of these cells to form large and smooth colonies decreased with time, demonstrating the typical behavior for cultures with mixed cell types comprising of KSCs, TACs, and DFCs (Figure 1c). Additionally, the number of cell doublings was not different among passages (Figure 1d); cultured HEKs underwent an average of 54.3 ± 1.9 doublings after ~50 days of culture (8 passages), with no signs of senescence.

### 2.2. Clonal Analysis 

Clonal analysis based on the study of clones derived from single cells gives crucial information about cell characteristics. In this study, 22% of keratinocytes were paraclones (DFCs) that formed very small and highly irregular colonies of large-sized cells. These clones proliferated for up to 20 cell generations over 25 days. The highest proportion of clones (73%) were meroclones (TACs), comprising an intermediate type of cell. The proliferation period of meroclones was limited to 52 cell generations over 48 days. The remaining 5% of keratinocytes were identified as holoclones (KSCs) that formed large colonies with a smooth and regular perimeter and small cell size (Figure 1e). Holoclones were able to sustain culturing for 60 days up to 67 cell generations, i.e., up to the end of this study (passage 8), with no signs of senescence (Figure 1f,g). These results demonstrate a higher proliferation capacity for holoclones compared to meroclones and paraclones in vitro.

### 2.3. Identification of Differentially Expressed Genes (DEGs)

Principal component analysis (PCA) showed that each type of colony clustered together with a clear separation between them (Figure 2a). The hierarchal cluster also separated the KSCs (holoclones) from their progenies (meroclones and paraclones). Meroclones and paraclones were clustered together; however, the meroclones were the first population to branch off, followed by the paraclones, which was in line with the direction of the differentiation process (Figure 2b).

To identify differentially expressed genes (DEGs), gene expression levels were compared between holoclones and meroclones, holoclones and paraclones, and between meroclones and paraclones. The analysis identified 1896 DEGs (upregulated: 943, downregulated: 953) between holoclones and meroclones; 2861 DEGs (upregulated: 1833, downregulated: 1028) between holoclones and paraclones, and 2893 DEGs (upregulated: 2007, downregulated: 886) between meroclones and paraclones (Figure 2c). The list of DEGs is shown in Appendix A.

### 2.4. Signature Genes and Gene Ontology

To identify specific markers, potential signature genes in each clonal type were investigated. For this purpose, a two-way Venn diagram was drawn from up- or downregulated genes using a specific clonal type as a baseline (Figure 2d). The analysis identified 301 and 149 unique up- and downregulated genes, respectively, in holoclones, 516 unique upregulated and 311 unique downregulated genes in meroclones, and 358 unique up- and 986 downregulated genes in paraclones (Appendix A). A list of the top 10 signature up- and downregulated genes are listed in Table 1. As expected, holoclones were enriched in KSC markers such as those encoded by *TP63*, *KRT15*, and *DLL1*. Meroclones (TACs) were enriched in genes (*THBS1, ASPM, MKI67, CCNA2, PTHLH, ASPM,* and *HMGA2*) related to the cell cycle and cell proliferation. Several markers related to the early- and late-stage of keratinocyte differentiation, such as those encoded by *KRT1*, *KRT10*, *FLG*, *IVL*, and *LOR,* were enriched in paraclones (DFCs). In addition to the expression of known genes, we also identified a set of novel genes (*JUNB, VEGFA, P63, K15, MXRA5, FAP*, *THSD1,* and *DCBLD2*) that could distinguish between KSCs and their progeny. These genes were confirmed either by quantitative real-time RT-PCR (qRT-PCR; *JUNB, VEGFA, P63, K15,* and *MXRA5*) or flow cytometry (*FAP*, *THSD1,* and *DCBLD2*; Appendix A).

To investigate the functional importance and biological processes associated with the signature DEGs in each colony type, the gene ontology (GO) of the signature up- and downregulated genes was analyzed separately. The results are shown in Appendix A. GO analysis was also used as a control of the separation method used in this study. As expected, holoclones were enriched in stem cells (SCs) related to GO terms, including regulation of transcription from RNA polymerase II promoter, cell proliferation, and the oxidation–reduction process (Figure 3). GO terms related to the cell cycle were enriched in meroclones (Figure 3). Those related to keratinocyte differentiation, including skin barrier establishment, keratinocyte differentiation, epidermis development, and innate immune response, were enriched in paraclones (Figure 3). Furthermore, we investigated additional important functional terms such as ‘cell surface’ (GO:0009986) from the cellular components category to identify cell surface genes that could differentiate the KSCs from their progeny. A total of 18 and 14 cell surface genes were enriched in KSCs and DFCs, respectively. We also identified eight genes in this category that were downregulated in KSCs (Table 2).

### 2.5. Signaling Pathway and Upstream Regulator Analysis

Ingenuity pathway analysis (IPA) of the 1869 DEGs between holoclones and meroclones identified 129 significantly altered canonical pathways either activated or inhibited; mainly, the EPK/MAPK signaling pathway, aldosterone signaling pathway, and cell cycle— *G1*/S cell cycle checkpoint regulation was activated in holoclones compared to meroclones. The IPA of the 2861 DEGs between holoclones and paraclones identified 137 significant canonical pathways, including EIF2 signaling and the NER pathway (all activated). A total of 63 significant canonical pathways, including the role of BRCA1 in DNA damage response, the NER Pathway, and purine nucleotide de novo biosynthesis II (all activated), were identified for the DEGs between meroclones and paraclones. The top activated/inhibited pathways for each clonal type compared to others are listed in Appendix A. In addition, a comparison of the DEGs between clonal types identified several activated and inhibited upstream regulators, including transcription factors (TFs; Figure 4 and Table 3). The top activated TFs identified between holoclones and meroclones included P53, FOS, FOXO1, EGFR, EGR1, VEGEA, H1F1A, HNF4A, DDIT3, ATF4, and TP63.

The top hub genes with the highest degrees in holoclones were *P53, FOS, FOXO1, EGFR, EGR1, VEGEA, H1F1A, HNF4A, DDIT3, ATF4*, and *TP63*. However, the cell cycle TFs (E2F1, E2F3, MYC, CCND1, TBX22, and FOXM1) were enriched in meroclones. The TFs related to cell cycle suppression, including RBL1, RB1, and CDKN2A, were found to be active in DFCs. The network nodes are proteins; the edges represent predicted functional associations. 

## 3. Discussion

KSCs rapidly lose their proliferative capabilities and differentiate early when cultured under inappropriate conditions [14], suggesting that the efficacy of the isolation and culturing protocol plays a critical role in the generation of a sufficient number of SCs and maintains their long-term proliferation potential in KSC culture. Several studies have compared different methods of isolation and maintenance of KSCs in vitro [6,15,16,17]. Taking these studies into account, irradiated 3T3-J2 fibroblast cells were used as a feeder layer to support HEK colony growth and maintain SC cultures in the present study. The clonogenic potential that represents the ability of a single cell to generate a colony of HEKs is an important parameter for assessing the quality of prepared cells [18]. Moreover, several studies have reported the relationship between clonogenic potential and stemness [6,19]. However, not all clonogenic cells can proliferate for longer cell generations. It has been shown that aborted colonies are generated by cells with very limited growth potential (less than 20 population doublings (PDs)), while large colonies can produce longer cell generations due to their higher self-renewal potential. Our results showed that the culture conditions and the use of serum-feeder layers were highly reproducible in enhancing the growth and proliferation rate of HEKs. Moreover, the cultured highly proliferative HEKs could generate colonies even after long-term culture and produce several cell generations (Figure 1c,d). These results are consistent with those of previous studies that used serum-feeder layer conditions [20,21]. 

Keratinocytes are characterized into holoclones, meroclones, and paraclones based on clonal analysis [6,18,21]. In this study, clonal analysis showed that most cells were meroclones (73%; TACs origin), while only 5% of cells were classified as holoclones (KSCs origin) with the highest proliferation capacity. In terms of percentages for each clonal type and the number of cell generations, these findings are in accordance with previous reports [20,22] and further confirm the efficacy of our culture protocol. 

Concordant with previous studies, the molecular characterization of these clonal types by microarray demonstrated that the expression levels of most promising KSC biomarkers, including MCSP [23], TP63 [22,24], DLL1 [25,26], SOX7 [27], KRT15 [28], ALDH1 [29], GBJ2 [30], and EGFR [31], were significantly upregulated in KSCs vs. TACs and DFCs (Appendix A). Furthermore, to assess the biological relevance of the signature DEGs in the KSCs, we focused on the genes implicated in other SC systems and investigated their possible role in regulating self-renewal and SC proliferation. Interestingly, we identified several SC-related signature genes including phosphoglycerate dehydrogenase (*PHGDH*), phosphoserine aminotransferase 1 (*PSAT1*), L-3-phosphoserine phosphatase (*PSPH*), thioredoxin-interacting protein (*TXNPI*), argininosuccinate synthetase (*ASS1*), *KRT23*, and caspase 14 (*CASP14*). It is known that serine is needed to synthesize proteins and other biomolecules involved in cell proliferation [32]. In this study, the enrichment of *PHGDH, PSAT1,* and *PSPH* genes that encode the key enzymes regulating the serine biosynthesis pathway in KSCs indicated the involvement of serine and the glycerin biosynthesis I pathway [33] in the proliferation of these cells. It has been reported that serine/glycine is a requirement of human muscle SC proliferation [34]. Furthermore, this pathway was also predicted by IPA (Appendix A). Taken together, it is evident that activation of serine and the glycerin biosynthesis I pathway is essential for HEK proliferation. A previous study has reported that *PSAT1* is essential for embryonic stem cell self-renewal and pluripotency [35]. However, our study is the first to show the role of *PSAT1* in epidermal SC proliferation. *TXNPI,* also known as vitamin D3 upregulated protein-1 or thioredoxin-binding protein-2, functions as an endogenous inhibitor of thioredoxin. It inhibits the antioxidative function of thioredoxin by binding to its redox-active cysteine residues [36,37,38]. Reportedly *TXNIP* regulates p53 and maintains hematopoietic cells by regulating intracellular ROS during oxidative stress [37]. The present study demonstrated that *TXNIP* might contribute, in part, to SC quiescence maintenance, similar to its role in the maintenance of human limbal epithelial basal cell quiescence through G0/G1 cell cycle arrest via p27kip1 [38]. A recent study using single-cell sequencing has shown that *ASS1* is expressed in the basal layer of IFE [30]. These results are in line with the current study. Moreover, ASS1 has been identified as the key enzyme required for de novo arginine generation, assuming that basal keratinocytes act as a source of cutaneous arginine, the “master and commander” of 21 innate immune responses [39]. Interestingly, in addition to the normal expression of *KRT15* in the basal layer, another keratin, *KRT23*, was enriched in our isolated KSC population. Recently, *KRT23* has been reported as a cancer stem cell (CSC) marker; however, its function in the skin is still unknown [40]. Another interesting finding was the expression of the *CASP14* gene, which was highly upregulated in KSCs and showed 51- and 4-fold higher expression in TACs and DFCs, respectively. *CASP14* is a non-apoptotic caspase involved in keratinocyte terminal differentiation and is important for skin barrier formation [41]. In contrast to our results, Gkegkes et al. has demonstrated that *CASP14* is predominantly expressed in highly differentiated layers [42]. Furthermore, a recent study demonstrated that *CASP14* is associated with CSC marker expression [43]. Collectively, these findings suggest that besides its role in cornification, *CASP14* could play an additional role in the keratinocyte developmental process. The study also demonstrated the unique upregulation of a few embryonic and adult stemness genes (*VEGFA, FGFR3, WEE1, ALDH2, RBL2, VANGL2, GJA1, EXPH5, USP3, FOPNL, ID1,* and *H19*) in KSCs. Overall, the present study identified several candidate KSC biomarkers; however, further investigations are required to elucidate the function of these biomarkers. 

Our findings also revealed that the DFCs were enriched with several genes associated with cell adhesion (DSC1, ADAM9, APBA1, APP, CD36, CD151, CTGF, CYR61, DDR2, FN1, ICAM1, INPPL1, ITGB6, L1CAM, LAMB3, MIR4260, NID2, SLAMF7, TINAGL1, and VCL). These genes play important roles in the adhesion of cells to an underlying substrate via cell adhesion molecules and adhesion of neighboring cells by the formation of intercellular adhesion structures such as tight junctions, adherence junctions and desmosomes [44]. For instance, DSC1 is strictly confined to suprabasal layers of the epidermis, but it is absent in mitotically active basal keratinocytes, which is in line with a previous study [45]. In addition, the expression levels of genes (ADAM17, ARF6, BCAM, CDH13, DSC3, EMP2, HES1, LOXL2, LY6D, MPZL3, PRKCA, PTPRS, PVRL1, TPBG, MXRA5, and MSLN) related to cell adhesion were also enriched in KSCs in the present study; however, their expression levels were different. These genes are involved in the regulation of cell adhesion, cell–cell adhesion and cell–substrate adhesion to anchor KSCs to the basement membrane and thus aid in KSC maintenance and prevent aging of SC [46,47]. For instance, MXRA5 encoding an adhesion proteoglycan belongs to a group of genes involved in extracellular matrix remodeling and cell–cell adhesion [48]. In our study, this gene was upregulated in KSCs by approximately 5-fold compared to TACs and DFCs (Table 1). Recently this gene has also been shown to be upregulated in mesenchymal stem cells (MSCs) [49]. MXRA5 gene expression was further validated by qRT-PCR, which showed similar results. The EMP2 gene encodes a protein that regulates cell membrane composition and is associated with various functions, including endocytosis, cell signaling, cell proliferation, cell migration, cell adhesion, and cell death. A previous study has shown that EMP2 is highly expressed in undifferentiated ESCs and MSCs [50]. On the contrary, a different set of genes (ANLN, ARHGAP18, ATIC, CAPZA1, CNN3, DIAPH3, FMNL2, HIST1H3J, HSPA8, SNORD14C, KIAA1524, RARS, RUVBL1, SNX5, and ZC3H15) related to cell–cell adhesion were upregulated in TACs. Furthermore, some of the genes (LY6D, BCAM (CD239), MARX5, PVRL1 (CD111), and MSLN) known to be differentially expressed in different clones were also present in the basal layer. These novel findings provide insights into the understanding of epidermal structure and function and pave the way for focused research in these areas.

When the surface markers for KSCs were explored in the present study, several cell surface markers were detected at high levels in KSCs compared to TACs and DFCs, including Galectin 1 (*LGALS1), LGALS7, CD68, CD109,* and *SLC7A11* (Table 2). Galectins are a family of β-galactoside-binding proteins implicated in SC and TAC behavior by modulating cell–cell and cell–matrix interactions. Two members of this gene family (*LGALS1* and *LGALS7)* were identified in our data set. *LGALS1* has been speculated to regulate apoptosis, cell proliferation, and cell differentiation, whereas *LGALS7* is mainly expressed in stratified squamous epithelia and has been shown to play a role in corneal epithelial cell migration and re-epithelialization of corneal wounds [51]. Another cell surface marker enriched in KSCs was *CD68*, a well-known marker for monocytes and macrophages. However, recent studies demonstrated that *CD68* is also expressed in MSCs derived from bone marrow, adipose tissue, and umbilical cord [52,53]. Interestingly, this is the first study to report its overexpression in KSCs compared to their progeny. In addition, CD109, a glycosylphosphatidylinositol (GPI)-anchored protein that inhibits TGF-β signaling, negatively regulates keratinocyte proliferation, and plays a role in the normal development of skin [54], was also enriched in KSCs. The cell surface markers *SLC1A4, SLC3A2*, and *SLC7A11* are the cell surface markers associated with amino acid transport. *SLC7A11* mediates cysteine uptake and glutamate release to promote glutathione synthesis, subsequently protects cells from oxidative stress and maintains the redox balance of cells, and, therefore, prevents lipid peroxidation-induced cell death [55]. Moreover, several studies have shown that *SLC7A11* is highly expressed in CSCs, and its knockdown attenuates their viability [56]. Additionally, among the 18 upregulated cell surface genes in KSCs, 10 have a role in regulating cell proliferation and cell growth (*ADAM17, EGFR, EMP2, FGFR3, SFRP1, VEGFA, SLC3A2, THBD, SLC7A1,* and *CEL*). Moreover, knockouts of some marker genes (e.g., transferrin receptor (*TFRC*)) have also been used to segregate KSCs from TACs [57]. In this study, we confirm the high expression of *TFRC* in TACs compared to KSCs and DFCs. Based on this, a novel set of negative cell surface markers, including *FAP, THSB1,* and *DCBLD2,* were identified (Table 2). The identified surface markers have different biological functions, including epithelial cell migration (*SLIT2, FAP, THBS1*), protein maturation (*CD59, PLAT, THSB1*), wound healing (*FAP, PLAT, THSB1*), apoptotic processes (*SLIT2, FAP, THBS1, PLAT*), vesicle coating (*AREG, CD59*), and negative regulation of cell growth (*CD59, DCBLD2*). Of these, *DCBLD2*, *FAP,* and *THBS1* were confirmed using flow cytometry (Appendix A). Overall, the novel information generated here could serve as a useful resource for selecting positive or negative cell surface markers to isolate KSCs. However, more studies are required to validate their function in the skin. 

Interestingly, genes associated with the GO terms “negative and positive regulation of transcription” were significantly found to be upregulated in KSCs (Appendix A). TFs are involved in development, and SC functions such as regulating cell fate determination, cell cycling, cell differentiation, and responses to the environment were also identified. Moreover, we used the upstream regulator analysis (URA) tool—part of the IPA software package—to identify potential upstream regulators based on the DEGs identified in the canonical pathway analysis (Figure 4). The PPI network analysis revealed that the top hub genes with the highest degrees in KSCs were *P53, FOS, FOXO1, EGFR, EGR1, VEGEA, H1F1A, HNF4A, DDIT3, ATF4*, and *TP63*. IPA predicted *P53* to be activated in KSCs and DFCs compared to TACs, suggesting its role in self-renewal and apoptosis. A recent study showed that *P53* preserves the proliferative potential of the SC compartment and limits the power of proto-oncogene MYC to drive cell cycle stress and differentiation [58]. Furthermore, *TP63* is crucial in the formation of stratified epithelium during epidermal development and in regulating epidermal tissue renewal [59,60]. In our study, *TP63* was upregulated in KSCs, which is in accordance with previous reports [22,61]. *TP63* expression was further validated by qRT-PCR and showed similar results (Appendix A). AP1 TFs are the most interesting and important regulators in the epidermis, which play key roles in regulating cell proliferation, apoptosis, and differentiation [62]. AP-1(JUN) TF consists of homo- or heterodimer members of the FOS (c-Fos, FosB, Fra-1, and Fra-2) and JUN (c-Jun, JunB, and JunD) proteins [63]. In our study, *FOS* and *JUNB* were upregulated in KSCs by 2.5- and 3-fold, respectively. Increased AP1 TF levels have been reported to be regulated by the MAPK signaling pathway [62]. Interestingly, 22 genes involved in the MAPK pathway were upregulated in KSCs, including *EGFR, TGFB1, NRAS, PAK1,* and *MAP3K1*. FOXO1 is another TF identified in KSCs. Several studies have shown that FOXO controls the self-renewal of hematopoietic, neural, and induced ESC populations, primarily by providing resistance to oxidative stress [64,65,66]. Additionally, FOXO TFs are involved in the regulation of cell differentiation [67,68]. Moreover, other studies on HEKs showed that FOXO1 is involved in the regulation of connective tissue wound healing and re-epithelization mediated via TGFB1 in vitro [69]. Hypoxia and hypoxia-inducible factors (HIFs) contribute to the maintenance of ESCs, the generation of induced pluripotent SCs, the functionality of hematopoietic SCs, and the survival of leukemia SCs [70]. Our study identified the activation of the TFs HIF1A, HIF2A, and HNF4A in KSCs, suggesting their role in maintaining KSCs potency. A different set of TFs was found to be enriched in TACs; most of these TFs are related to cell cycle regulation, including E2F1, E2F3, MYC, CCND1, TBX22, and FOXM1. Furthermore, the TFs related to cell cycle suppression, including RBL1, RB1, and CDKN2A, were found to be active in DFCs (Figure 4). These findings are in line with the nature of TACs being more rapidly cycling, whereas DFCs possess limited proliferation potential. 

## 4. Materials and Methods

### 4.1. Ethical Statement and Informed Consent 

The study was approved by the Institutional Review Board at the Cell Therapy Center/The University of Jordan (approval code: IRB/1/2015, date: 19 February 2015). The experiments were conducted following the principles outlined in the Declaration of Helsinki for all human experimental investigations.

### 4.2. Isolation and Culture of Human Epidermal Keratinocytes (HEKs)

HEKs were isolated from neonatal foreskin samples obtained from healthy newborns (*n* = 6) after circumcision following a method described previously [4,32]. Briefly, the tissue was cut into small pieces and trypsinized using 0.05% Trypsin–EDTA for 2 h at 37 °C with continuous agitation. Cells were collected every 30 min and counted using trypan blue (All from Gibco, Thermo Fisher Scientific, USA). Cells were cultured at a seeding density of 2.5 × 10^4^/cm^2^ on a feeder layer of lethally irradiated 3T3-J2 cells. Complete keratinocyte media [DMEM and Ham’s F12 media (2:1 mixture), 10% FCS, glutamine (4 mM), penicillin–streptomycin (50 IU/mL; all from Gibco), adenine (0.18 mM), insulin (5 μg/mL), cholera toxin (0.1 nM), triiodothyronine (2 nM), and hydrocortisone (0.4 μg/mL; all from Sigma Aldrich, St Louis, MO, USA)) were added to each well. Epidermal growth factor (10 ng/mL; Austral Biologicals, San Ramon, CA, USA) was added to the culture media, starting on day 3. The media were changed every alternative day. Subconfluent primary cultures were passaged at a density of 6 × 10^3^ cells/cm^2^ and cultured as above. All cultures were incubated in 5% CO_2_ at 37 °C.

### 4.3. Characterization of Keratinocyte Cultures

#### 4.3.1. Immunofluorescence Staining

For the observation of cell morphology and identification of HEKs in culture, immunofluorescence staining of cytokeratin 5, cytokeratin 14, and nuclei were performed on primary cell culture. In brief, cells were fixed using 4% (*w*/*v*) paraformaldehyde for 15 min and permeabilized by 0.25% (*v*/*v*) Triton X-100 in PBS for 30 min, followed by blocking at room temperature (RT) for 1 h with BlockAid™ Blocking Solution (Thermo Fisher Scientific). The cells were incubated in primary antibodies mix (Anti-K14 (rabbit monoclonal/IgG) and Anti-K5 (mouse monoclonal/IgG1)) at 4 °C overnight. After washing with PBS, the secondary antibodies (Alexa Fluor 488-goat anti-mouse and Alexa Fluor 594-goat anti-rabbit) were added to the samples for 1 h at RT (all antibodies were from Abcam, Cambridge, MA, USA). The cells were counterstained with 4′,6-diamidine-2′-phenylindoledihydrochloride (DAPI; Thermo Fisher Scientific) for 10 min and visualized using a fluorescence microscope (Zeiss, Jena, Germany). 

#### 4.3.2. Colony-Forming Efficiency

Colony-forming Efficiency (CFE) was performed to indicate the capacity of a basal cell to generate a colony. 1000 cells from each biopsy and cell passage were plated onto 3T3-J2 feeder layers and cultured as described above. Twelve days later, the colonies were fixed and stained with 2% Rhodamine B following the method described in a previous study [32]. The experiments were performed in duplicate. Colony-forming efficiency (CFE) values were calculated using the following formula:CFE (%) = (Colonies Counted/Cells Inoculated) × 100

#### 4.3.3. Proliferation Potential

Proliferation potential was assessed to indicate the capacity of the isolated cells to produce cell generations. HEKs were continuously cultured on the 3T3-J2 feeder layer up to passage 8 (~50 days). At each passage, cells were trypsinized and counted using trypan blue and cultured at a seeding density of 6 × 10^3^ cells/cm^2^ for the next passage. The population doubling (PD) of cultured HEKs was calculated using the following equation [32]: PD = 3.322 log N/N_0_
where N is the number of harvested cells and N_0_ is the number of clonogenic cells. 

### 4.4. Clonal Analysis 

Clonal analysis was based on the study of the clones derived from single cells. It gives very important information about cell characteristics. Clonal analysis was used in this study to isolate different clonal types (holoclone, meroclone, and paraclone). Clonal analysis was performed using subconfluent HEKs from primary culture according to the method reported by Pellegrini et al. [32]. After the colonies reached 70–80% confluency, single cells were inoculated onto 96-well plates containing a 3T3-J2 feeder layer. Seven days later, colonies were identified, and 100 colonies were studied for each sample. Subsequently, each selected clone was trypsinized and transferred onto two dishes, 3/4 of the clone were used in (i) serial propagation (CFE assay and PD measurement as described above) and (ii) RNA extraction, while 1/4 of the clone was cultured on a 60 mm dish in standard culture conditions for clonal classification. To determine the clonal types, the colonies were stained after 12 days of culture as described above for clonal classification [6], and the percentage of aborted colonies (<3 mm^2^ in diameter) formed by the progeny of the founding cell was estimated using the following formula: Aborted colonies (%) = (Aborted colonies counted/Total colonies counted) × 100 

Based on the percentage of aborted colonies, they were classified as holoclone (aborted colonies: 0–5%), meroclone (aborted colonies: 5–95%), or paraclone (aborted colonies: >95%). 

### 4.5. Whole Transcriptome Analysis 

#### 4.5.1. RNA Extraction

RNA was extracted from six colonies from each clonal type (holoclone, meroclone, and paraclone). Cells were lysed using Trizol^®^ reagent (Ambion, ThermoFisher Scientific), and the total RNA was extracted using an RNeasy Mini kit (Qiagen, Valencia, CA, USA) following the manufacturer’s instructions. The total RNA yield and purity were measured using a NanoDrop 2000c spectrophotometer system (Thermo Fisher Scientific). RNA integrity was analyzed using an Agilent 2100 Bioanalyzer and Agilent RNA 6000 Nano Kit (Agilent, Santa Clara, CA, USA) following the manufacturer’s instructions. 

#### 4.5.2. Global Gene Expression Profiling

Total transcriptome analysis was performed for holoclones, meroclones, and paraclones, each from six different donors. Gene expression profiling of total RNA was performed using GeneChip^®^ Human Transcriptome Array 2 (HTA 2; Affymetrix. Inc., Santa Clara, CA, USA) according to the manufacturer’s instructions. Briefly, 100 ng of total RNA from each sample was amplified using a GeneChip^®^ WT PLUS Reagent Kit, and 5.5 µg of amplified cDNA from each sample was fragmented, labeled, and hybridized into the GeneChip HTA 2 chips. Following hybridization, the chips were washed and stained using an Affymetrix^®^ automated Fluidics station FS450. The chips were subsequently laser-scanned using an Affymetrix^®^ GeneChip Scanner 3000 7G. The microarray data can be accessed via the Gene Expression Omnibus (accession number GSE148164).

#### 4.5.3. Data Analysis

Affymetrix HTA2.0 chip.CEL files were normalized by a Signal Space Transformation-robust Multi-Array Analysis (SST-RMA) algorithm using Affymetrix Transcriptome Analysis Console (TAC) software version 4.0.1 (Affymetrix, Santa Clara, CA, USA). PCA and hierarchical cluster analysis were performed using TAC V4.0.1 software. Repeated measures ANOVA was performed to compare the expression change between the clones, with a statistical significance level set at *p* < 0.05. Experimental batch effects were adjusted by including an experimental batch as a covariate in our statistical model. Genes that exhibited a 1.5-fold change at *p* < 0.05 were filtered. IPA software (Ingenuity Systems, Redwood City, CA, USA) was used for pathway analyses. Additionally, selected genes were annotated using the Database for Annotation, Visualization, and Integrated Discovery (DAVID) v6.8 (https://david.ncifcrf.gov/home.jsp, accessed on 5 February 2021). TFs identified by DAVID and predicted by IPA software were mapped to the search tool to retrieve interacting genes using the STRING database (https://string-db.org/, accessed on 10 March 2021) and to acquire protein–protein interaction (PPI) networks. 

### 4.6. Validation of the Microarray Results 

#### 4.6.1. Quantitative Real-Time Polymerase Chain Reaction (qRT-PCR)

To verify gene expression data obtained from the HTA 2 assay, qRT-PCR for DEGs, including *JUNB, VEGFA, P63, K15,* and *MXRA5*, was performed. The primers were designed using the primer-blast tool at NCBI (Appendix A). The SuperScript VILO cDNA Synthesis Kit (Invitrogen, Carlsbad, CA, USA) was used to synthesize cDNA from 1 μg RNA according to the manufacturer’s instructions. qRT-PCR was performed using SYBR Premix Ex Taq II (Takara, Japan) and a ViiA7 real-time PCR system (Applied Biosystems, Foster City, CA, USA). Data were normalized against cyclophilin A levels, and the relative fold change was calculated using ViiA7 software (Life Technologies, Foster City, CA, USA). All reactions were performed in triplicate.

#### 4.6.2. Flow Cytometry

Expression of cell surface markers at the protein level was selectively analyzed by flow cytometry according to the microarray results. Holo-, mero-, and paraclones (*n* = 3, each) were harvested and suspended in PBS at a concentration of 1 million cells per mL. For the surface staining, anti-hTHSD1 (R&D systems #FAB3715P), with an appropriately matched isotype control mAb (R&D systems #IC002G), was used. The following antibodies were used for intracellular staining: anti-hFAP and anti-hDCBLD2 (R&D systems # FAB3715P, AF6269), with an appropriately matched isotype control mAb (R&D systems # IC002P, 5-001-A). The results were assessed using an FACScan (BD Biosciences, San Jose, CA, USA).

## 5. Conclusions

The ability to identify, purify, and characterize KSCs is a critically important issue in epidermal stem cell biology. Because of the absence of KSC-specific markers, cell size remains the most reliable and efficient means to enrich clonogenic keratinocytes as single cells. The present study unraveled the potential of single-cell-based colony characterization coupled with microarray technology. The findings established the “transcriptome,” a comprehensive databank of genes expressed in the KSCs and their progenies (TACs and DFCs). Furthermore, the differences in the expressions of the genes related to proliferation, adhesion, and TFs confirmed the effective isolation of the cell populations. These findings are considered a valuable source for identifying and characterizing molecular marker(s) in KSCs and their progeny. Significantly, such molecular markers would allow for the identification and isolation of stem cells and their specific use in clinical cultures. In addition, they could provide an evaluation method to improve the quality and success rate of epidermal grafts due to the increase in SC percentage. Furthermore, we believe that these findings will pave the way for exploring the precise mechanisms underlying the regulation and differentiation of KSCs.

## Figures and Tables

**Figure 1 ijms-22-10810-f001:**
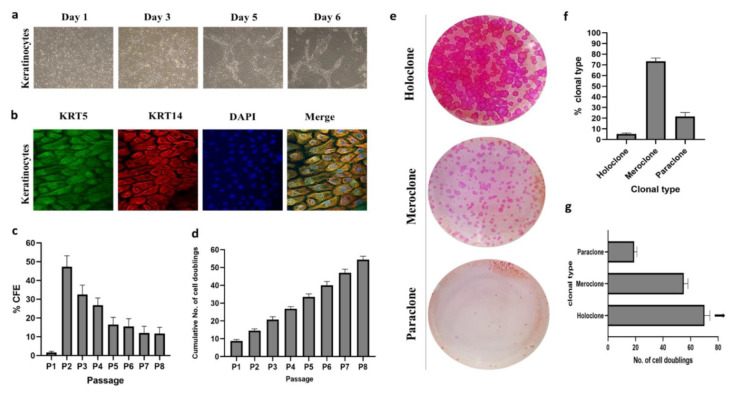
Characterization of human keratinocytes derived from foreskin. (**a**) Phase-contrast microscopic analysis for cell morphology of keratinocytes (P2) at different time points (40×). (**b**) Confocal images representing immunofluorescence shows the expression of epidermal markers KRT5 (green) and KRT14 (red), with the nucleus also stained blue using DAPI (100×). (**c**) The percentage of CFE and (**d**) cumulative number of cells doubling in each passage, which are representative of 6 samples, expressed as mean ± SD. (**e**) Clonal analysis and classification of clonal types; classification was based on the percentage of aborted colonies (<3 mm^2^ in diameter); when 0–5% of colonies were aborted the clone was scored as holoclone. When more than 95% of the colonies were aborted, the clone was classified as paraclone. When more than 5%, but less than 95% of the colonies were aborted, the clone was classified as meroclone. (**f**) Average percentage of holoclones, meroclones, and paraclones (*Y*-axis) were identified by clonal analysis of 6 donor samples. (**g**) The number of cell doublings performed by cultured cells, which were calculated using the following formula: x = 3.322 log N/N_o_, where N equals the total number of cells obtained at each passage and N_o_ equals the number of clonogenic cells plated. The arrows indicate cells that continued to divide after passage 8.

**Figure 2 ijms-22-10810-f002:**
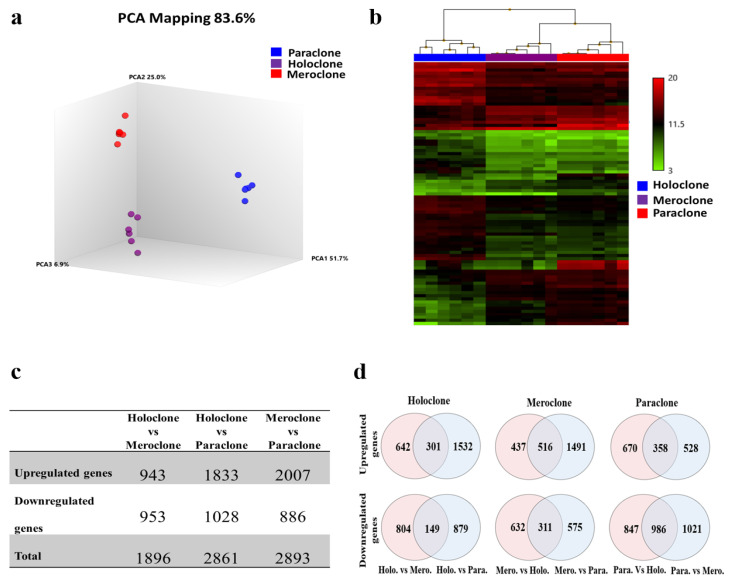
Snapshot microarray analysis of holoclones, meroclones, and paraclones. (**a**) Principal component analysis (PCA) of all the colony types profiled. Holoclones (indicated by purple color) were clustered distinctly from meroclones (indicated by red color) and paraclones (indicated by blue color). The percentage values indicate the proportion of total variance described by each PCA. PCA1 (*X*-axis); PCA 2 (*Y*-axis); PCA 3 (*Z*-axis). (**b**) The hierarchal cluster also separated the KSCs (holoclones) from their progenies (meroclones and paraclones). (**c**) Total number of differentially expressed genes for each clonal type compared to others. (**d**) Two-way Venn diagrams for specifically up and downregulated genes in holoclones, meroclones, and paraclones. The number in the middle of each of the two circles represents the number of specifically up- or downregulated genes in each clonal type.

**Figure 3 ijms-22-10810-f003:**
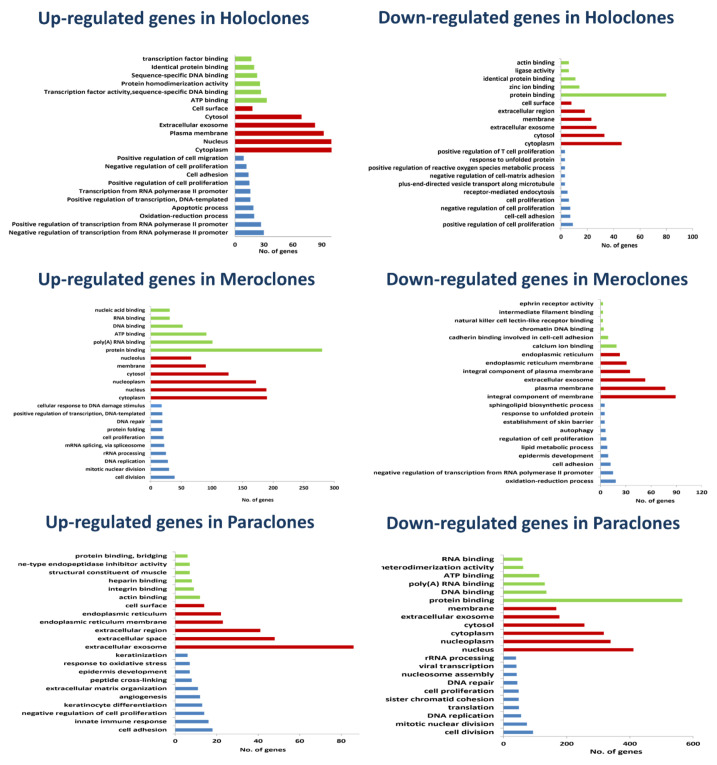
Gene ontology of the signature differentially expressed genes. The DEGs in holoclones, meroclones, and paraclones were annotated into three major functional categories: biological process (blue bars), cellular component (red bars), and molecular function (green bars). The *X*-axis indicates the names of GO terms (Direct level) and the *Y*-axis indicates the number of genes.

**Figure 4 ijms-22-10810-f004:**
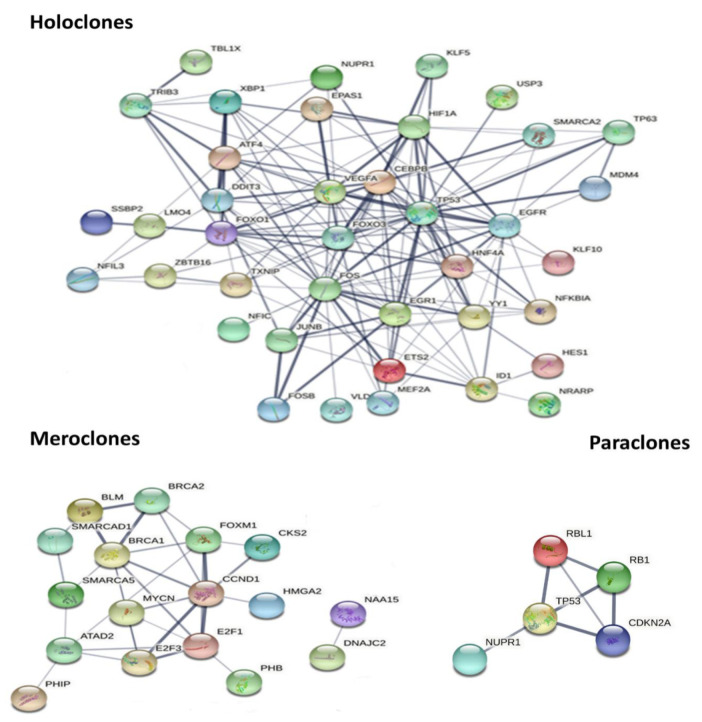
Protein–protein interaction (PPI) network based on STRING database of identified and predicted transcription factors.

**Table 1 ijms-22-10810-t001:** Top signature genes up- or downregulated uniquely in each clonal type.

Gene Symbol	Description	Holo vs. Mero FC	Holo vs. Para FC
Holoclone Upregulated Genes			
*CASP14*	caspase 14	50.69	3.85
*ASS1*	argininosuccinate synthase 1	25.53	6.51
*PSAT1*	phosphoserine aminotransferase 1	21.47	14.64
*H19; MIR675*	H19, imprinted maternally expressed transcript (non-protein coding); microRNA 675	21.11	32.67
*SLC7A11*	solute carrier family 7 (anionic amino acid transporter light chain, xc- system), member 11	16.75	19.82
*ASNS*	asparagine synthetase (glutamine-hydrolyzing)	10.58	6.26
*LGALS7; LGALS7B*	lectin, galactoside-binding, soluble, 7; lectin, galactoside-binding, soluble, 7B	9.64	2.11
*IMPA2*	inositol(myo)-1(or 4)-monophosphatase 2	9.57	5.86
*TXNIP*	thioredoxin interacting protein	9.32	1.89
*CHAC1*	ChaC glutathione-specific gamma-glutamylcyclotransferase 1	9.19	3.03
Holoclone Downregulated Genes			
*PTHLH*	parathyroid hormone-like hormone	−9.81	−4.33
*DCBLD2*	discoidin, CUB and LCCL domain containing 2	−9.18	−5.71
*THBS1*	thrombospondin 1	−8.71	−6.22
*DKK1*	dickkopf WNT signaling pathway inhibitor 1	−8.52	−4.72
*KRTAP2-3*	keratin associated protein 2–3	−6.28	−50.55
*TSPAN1*	tetraspanin 1	−5.6	−6.74
*PLAT*	plasminogen activator, tissue	−5.41	−9.5
*FAP*	fibroblast activation protein alpha	−5.32	−2.45
*ENC1*	ectodermal-neural cortex 1 (with BTB domain)	−4.87	−7.82
*ADAMTS1*	ADAM metallopeptidase with thrombospondin type 1 motif 1	−4.8	−12.59
Meroclone Upregulated Genes			
*FST*	follistatin	6.81	5.42
*RNU6-26P*	RNA, U6 small nuclear 26, pseudogene	3.95	1.89
*TFRC*	transferrin receptor	3.88	2.8
*SMURF2*	SMAD specific E3 ubiquitin protein ligase 2	3.73	2.14
*ADAMTS6*	ADAM metallopeptidase with thrombospondin type 1 motif 6	3.64	2.24
*HMGA2*	high mobility group AT-hook 2	3.48	3.28
*CENPE*	centromere protein E	3.09	6.5
*C20orf197*	chromosome 20 open reading frame 197	3.08	2.8
*CLSPN*	claspin	3.02	4.74
*UTP20*	UTP20 small subunit (SSU) processome component	2.98	2.53
Meroclone Downregulated Genes			
*CASP14*	caspase 14	−50.69	−13.15
*ALOX12B*	arachidonate 12-lipoxygenase, 12R type	−13.52	−26.39
*ATP12A*	ATPase, H+/K+ transporting, nongastric, alpha polypeptide	−10.69	−18.9
*C10orf99*	chromosome 10 open reading frame 99	−10.26	−6.47
*LGALS7; LGALS7B*	lectin, galactoside-binding, soluble, 7; lectin, galactoside-binding, soluble, 7B	−9.64	−4.58
*TXNIP*	thioredoxin interacting protein	−9.32	−4.93
*CHAC1*	ChaC glutathione-specific gamma-glutamylcyclotransferase 1	−9.19	−3.04
*LGALS7B*	lectin, galactoside-binding, soluble, 7B	−8.7	−4.14
*SCNN1B*	sodium channel, non voltage gated 1 beta subunit	−7.77	−8.19
*CDHR1*	cadherin-related family member 1	−7.51	−4.02
Paraclone Upregulated Genes			
*FLG*	filaggrin	269.82	86.31
*KRT1*	keratin 1, type II	156.48	36.68
*LY6G6C*	lymphocyte antigen 6 complex, locus G6C	41.26	15.44
*CRNN*	cornulin	24.83	23.29
*SERPINB12*	serpin peptidase inhibitor, clade B (ovalbumin), member 12	22.69	6.67
*THEM5*	thioesterase superfamily member 5	19.34	9.05
*KRT10*	keratin 10, type I	18.75	9.74
*CEACAM5*	carcinoembryonic antigen-related cell adhesion molecule 5	13.64	3.58
*PTPRM*	protein tyrosine phosphatase, receptor type, M	10.45	4.36
*MIR3687-1*	microRNA 3687-1	9.93	6.15
Paraclone Downregulated Genes			
*CCNA2*	cyclin A2	−20.09	−13.53
*MKI67*	marker of proliferation Ki-67	−17.62	−13.92
*KIF11*	kinesin family member 11	−15.14	−8.72
*DLGAP5*	discs, large (Drosophila) homolog-associated protein 5	−14.97	−8.49
*HIST1H3B*	histone cluster 1, H3b	−14.48	−12.24
*NUSAP1*	nucleolar and spindle associated protein 1	−14.14	−11.9
*TOP2A*	topoisomerase (DNA) II alpha	−13.94	−10.46
*CENPF*	centromere protein F	−12.41	−9.44
*NCAPG*	non-SMC condensin I complex subunit G	−10.75	−5.57
*HIST1H3I*	histone cluster 1, H3i	−10.32	−9.18

**Table 2 ijms-22-10810-t002:** List of the gene expression levels of cell surface markers revealed by microarray analysis.

Gene Symbol	Description	Holo vs. Mero FC	Holo vs. Para FC	Colony F-Test
	Holoclones			
*SLC7A11*	solute carrier family 7 (anionic amino acid transporter light chain, xc- system), member 11	16.7	19.8	3.55 × 10^−7^
*CEL*	carboxyl ester lipase	2.5	6	0.0052
*VEGFA*	vascular endothelial growth factor A	4	4	5.64 × 10^−6^
*LGALS1*	lectin, galactoside-binding, soluble, 1	3.3	10	1.39 × 10^−5^
*CD68*	CD68 molecule	3	3.8	4.10 × 10^−3^
*THBD*	thrombomodulin	3.2	1.5	0.001
*SLC3A2*	solute carrier family 3 (amino acid transporter heavy chain), member 2	2.9	2.2	0.0004
*CD109*	CD109 molecule	2.11	2.3	0.048
*MSLN*	mesothelin	2.5	4	0.0234
*PLA2R1*	phospholipase A2 receptor 1	2	1.5	0.0112
*EMP2*	epithelial membrane protein 2	2.2	2.4	1.39 × 10^−5^
*WNT4*	wingless-type MMTV integration site family, member 4	1.8	1.9	2.99 × 10^−5^
*ADAM17*	ADAM metallopeptidase domain 17	2	1.7	0.0003
*SFRP1*	secreted frizzled-related protein 1	1.69	2.1	7.75 × 10^−5^
*FGFR3*	fibroblast growth factor receptor 3	2.2	2.6	0.0011
*EGFR*	epidermal growth factor receptor	1.6	1.7	0.0052
*SLC1A4*	solute carrier family 1 (glutamate/neutral amino acid transporter), member 4	2.4	1.73	0.0102
*BACE1*	beta-site APP-cleaving enzyme 1	2.8	1.5	3.00 × 10^−5^
*CD59; C11orf91*	CD59 molecule, complement regulatory protein; chromosome 11 open reading frame 91	−1.5	−1.7	0.0078
*PLAT*	plasminogen activator, tissue	−3.8	−6	0.0004
*AREG*	amphiregulin	−2.2	−2	0.001
*TFRC*	transferrin receptor	−3.9	−1.77	0.0048
*SLIT2*	slit guidance ligand 2	−2.8	−1.6	0.0018
*FAP*	fibroblast activation protein alpha	−5.3	−2.5	0.0039
*THBS1*	thrombospondin 1	−8.7	−6.2	0.0002
*DCBLD2*	discoidin, CUB and LCCL domain containing 2	−9.2	−5.7	8.94 × 10^−5^
	Paraclones			
*TNFSF18*	tumor necrosis factor (ligand) superfamily, member 18	1.57	1.55	0.0005
*PROM2*	prominin 2	3.62	3.13	0.0154
*ICAM1*	intercellular adhesion molecule 1	1.68	1.52	0.0394
*HLA-H*	major histocompatibility complex, class I, H (pseudogene)	1.67	2.14	0.0059
*HLA-C*	major histocompatibility complex, class I, C	2.2	2.63	0.0049
*HLA-B*	major histocompatibility complex, class I, B	1.88	2.28	0.0185
*HLA-A*	major histocompatibility complex, class I, A	2.41	3.8	0.0005
*CPM*	carboxypeptidase M	1.97	4.58	0.0035
*CLU*	clusterin	1.62	1.52	0.013
*CEP295NL; TIMP2*	CEP295 N-terminal like; TIMP metallopeptidase inhibitor 2	1.65	1.64	0.0333
*CD36*	CD36 molecule (thrombospondin receptor)	1.85	1.93	0.0237
*ANXA9*	annexin A9	1.77	3.2	0.031
*AMOT; MIR4329*	angiomotin; microRNA 4329	3.31	2.81	0.0006
*ACE2*	angiotensin I converting enzyme 2	1.5	1.55	0.0123

**Table 3 ijms-22-10810-t003:** Transcription regulators predicted to be activated/inhibited by ingenuity pathway analysis (IPA).

Upstream Regulator	Predicted Activation State	Activation Z-Score
Holoclone vs. Meroclone		
MYC	Inhibited	−2.5
ATF4	Activated	3.9
TP53	Activated	3.6
TP63	Activated	1.7
HNF4A	Activated	2.7
NUPR1	Activated	6
HIF1A	Activated	2.8
EPAS1	Activated	2
CCND1	Inhibited	−2.7
Meroclone vs. Paraclone		
E2F1	Activated	4.8
TP53	Inhibited	−5.5
MYC	Activated	8.5
CCND1	Activated	3.8
E2F3	Activated	6.2
TBX2	Activated	4.8
NUPR1	Inhibited	−9.5
CDKN2A	Inhibited	−7.8
RB1	Inhibited	−3.8
FOXM1	Activated	5.8
E2F2	Activated	3
RBL1	Inhibited	−4.5
MAX	Activated	2.4
Holoclone vs. Paraclone		
TP53	Inhibited	−4.7
MYC	Activated	7.6
E2F1	Activated	4.8
CDKN2A	Inhibited	−7.1
RB1	Inhibited	−4.1
NUPR1	Inhibited	−6.5
CCND1	Activated	3.4
TBX2	Activated	4.3
E2F3	Activated	4.7
FOXM1	Activated	5.5
RBL1	Inhibited	−4.9
EP400	Activated	4.8
MYCN	Activated	6.7
E2F2	Activated	3

## Data Availability

The data that support the findings of this study have been deposited in the Gene Expression Omnibus (https://www.ncbi.nlm.nih.gov/geo/, accessed on 6 April 2020) with accession number GSE148164. All data analyzed during this study have been included in this published article and its Appendix A.

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
