# Peer review of "Differential Marker Expression between Keratinocyte Stem Cells and Their Progeny Generated from a Single Colony"

_ijms, 2021, doi:10.3390/ijms221910810_

Round 1

Reviewer 1 Report

  Thank you very much for submitting your Manuscript ID ijms-1393482 “ Differential marker expression between keratinocyte stem cells and their progeny generated from a single colony” for the International Journal of Molecular Science.  I am surprised at big difference of transcriptomic analysis regarding three type of cell size in KSC. This manuscript blows a whistle raise for researchers to perform the transcriptomic analysis using various primary cells.  I think this manuscript is accepted soon.  However, I have a few concerns of this one.  I wish the authors will revise in accordance with my suggestion.  Thank you.

  • Tables are too much. The duplicate data should be deleted or be moved to supplemental Materials. I guess it is not necessary for Table 1, Table 2 or Fig. 3, and Table 3 or Table 4.
  • At least, Figure 2 b is too small and be deleted.
  • I cannot find the validated data by qPCR and FC analysis. Please show compared among three methods in Table or Figure.

Reviewer 2 Report

The study is well designed, perfored and illustrated and the results seem to be valid. However, I would suggest the authors to better explain the readers the idea of the study and the difference of different potential of keratinocytes to form the collonies. The aurthors should also better explain the relevance of their aim and findings.

Author Response

Comments and Suggestions for Authors

The study is well designed, performed and illustrated and the results seem to be valid. However, I would suggest the authors to better explain the readers the idea of the study and the difference of different potential of keratinocytes to form the colonies. The authors should also better explain the relevance of their aim and findings.

Our answer: We are pleased about the favorable opinion of the reviewer about our work, and we want to thank the reviewer for the valuable comments that gave us the chance to improve the manuscript strongly.

The introduction of the paper has been modified in which we better explain the differences in proliferation capacities for different keratinocytes cell types (lines 39-47). In addition, the relevance of the aims and findings were also further explained first in the introduction and was also included in more detail in the conclusions (lines 612-627).

Reviewer 3 Report

Ali et al. studied the differentially expressed genes in keratinocyte stem cells for potential stem cell-specific markers. The study is interesting, and the result is potentially informative. However, minor corrections are recommended to improve clarity and readability before publication. Specific comments are listed below:

  • It would be very helpful to have an illustration explaining the anatomy, biology, and the overall mindset behind the experiment, more like an expanded high-level abstract figure for general readers from outside of the field. The manuscript dives right into technical details with insufficient background knowledge introduction/illustration in the present format. The same applies to the description of the methodology. The definition for each measurement needs to be better explained within this manuscript. Figure 4 is cropped and incomplete. The green in Figure 3d is also different from others (does that mean anything?).

  • The visibility of the figures has to be improved. For example, it is difficult to navigate Figure 2a-b and difficult to read Figure 3 and Figure 4 with the small font size and blurred scripts. Figure 2-4 also needs to improve the captions to explain the data to guild the reader, followed by clear conclusions from each figure. So far, the captions are mainly plain descriptions/annotations with insufficient context.  

  • PC-3 only accounts for 6.9%. Therefore, if the data is well separated by PC-1 and PC-2, it is best to plot just PC-1 and PC-2 in a 2D graph instead of having it in 3D.

  • The table is very hard to read with horizontal lines for every row and hard to find the headers for each data block. Use only top and bottom lines for each block of data will be sufficient.

  • The conclusion section is separated from the discussion with methods in-between. It reads interrupting. The discussion/results centering Fig 4 is also very difficult to read. Alternatives in data visualization have to be considered to improve clarity (similar comment for the introduction section). After all, although well-intended, no matter how informative the study is, the manuscript is only as useful/helpful/impactful as how it is understood/read by the general readers.

Author Response

Comments and Suggestions for Authors

Ali et al. studied the differentially expressed genes in keratinocyte stem cells for potential stem cell-specific markers. The study is interesting, and the result is potentially informative. However, minor corrections are recommended to improve clarity and readability before publication. Specific comments are listed below:

  • It would be very helpful to have an illustration explaining the anatomy, biology, and the overall mindset behind the experiment, more like an expanded high-level abstract figure for general readers from outside of the field. The manuscript dives right into technical details with insufficient background knowledge introduction/illustration in the present format.

Our answer: We are grateful for the reviewer's comments to help us to improve on the manuscript. Following critical re-reading of the introduction and methodology, we agree with the reviewer. The introduction dives deep into the science behind the research described without adequate background and clinical/application significance. We have reorganized the introduction to address these concerns and now believe it reads better to a wider audience.

  • The same applies to the description of the methodology. The definition for each measurement needs to be better explained within this manuscript.

Our answer: An explanation including the purpose of each methodology used in this study was included in the manuscript. (Lines 507-509) (Lines 521-522)

  • Figure 4 is cropped and incomplete. The green in Figure 3d is also different from others (does that mean anything?).

Our answer: We thank the reviewer for his thorough observation. Figure 4 is checked and found to be complete. The green color for figure 3d was overlooked from our side. However, we modified figure 3d colors to match the rest of figure 3 parts.

  • The visibility of the figures has to be improved. For example, it is difficult to navigate Figure 2a-b and difficult to read Figure 3 and Figure 4 with the small font size and blurred scripts. Figure 2-4 also needs to improve the captions to explain the data to guild the reader, followed by clear conclusions from each figure. So far, the captions are mainly plain descriptions/annotations with insufficient context.  

Our answer: we have modified the figures font to be more readable and clear for the readers and modified the captions of the figures to explain the data better. Changes in figures captions are tracked in the main manuscript 

  • PC-3 only accounts for 6.9%. Therefore, if the data is well separated by PC-1 and PC-2, it is best to plot just PC-1 and PC-2 in a 2D graph instead of having it in 3D.

Our answer: we thank the reviewer for this observation. However, we included the PC-3 in our figure to allow for more precise visualization of the separation of the three clonal types of cells. 

  • The table is very hard to read with horizontal lines for every row and hard to find the headers for each data block. Use only top and bottom lines for each block of data will be sufficient.

 Our answer: We are grateful for the reviewer's comment to help us improve the manuscript. The tables have been modified by removing the horizontal lines for every row to make the table easier to read and follow.

  • The conclusion section is separated from the discussion with methods in-between. It reads interrupting. The discussion/results centering Fig 4 is also very difficult to read. Alternatives in data visualization have to be considered to improve clarity (similar comment for the introduction section). After all, although well-intended, no matter how informative the study is, the manuscript is only as useful/helpful/impactful as how it is understood/read by the general readers.

Our answer: We agree with the reviewer that the conclusion reads interrupting after the methods. However, it was placed after the methods according to the journal instruction. In addition, the discussion/ result section for figure 4 was modified for easier and better readability (lines 264-279 ) (lines 477-484)